# Curcumin-like Compound Inhibits Proliferation of Adenocarcinoma Cells by Inducing Cell Cycle Arrest and Senescence

**DOI:** 10.3390/ph18060914

**Published:** 2025-06-18

**Authors:** Rafael Fonseca, Yasmin dos Santos Louzano, Cindy Juliet Cristancho Ortiz, Matheus de Freitas Silva, Maria Luiza Vieira Felix, Guilherme Álvaro Ferreira-Silva, Ester Siqueira Caixeta, Bruno Zavan, Claudio Viegas, Marisa Ionta

**Affiliations:** 1Laboratório de Avaliação de Protótipos Antitumorais (LAPAN), Instituto de Ciências Biomédicas, Universidade Federal de Alfenas (UNIFAL-MG), Alfenas 37130-001, Brazil; rafael.miranda@sou.unifal-mg.edu.br (R.F.); yasmin.louzano@sou.unifal-mg.edu.br (Y.d.S.L.); maria.felix@sou.unifal-mg.edu.br (M.d.F.S.); maria.luiza@sou.unifal-m.edu.br (M.L.V.F.); guilherme.ferreira@unifal-mg.edu.br (G.Á.F.-S.); ester.caixeta@unifal-mg.edu.br (E.S.C.); brunozavan@yahoo.com.br (B.Z.); 2Laboratory of Research in Medicinal Chemistry (PeQuiM), Institute of Chemistry, Federal University of Alfenas, Alfenas 37133-840, Brazil; cjcristanchoo@unal.edu.co

**Keywords:** non-small cell lung cancer, curcumin-like compounds, cell cycle arrest, senescence

## Abstract

**Background:** Lung cancer is the leading cause of cancer-related death in the male sex worldwide. Non-small cell lung cancer (NSCLC) is the most prevalent type, accounting for 80–85% of cases, and lung adenocarcinoma is the most common and lethal NSCLC subtype, being responsible for ca. 50% of deaths. Despite new therapeutic strategies, lung cancer mortality rates remain high, highlighting the need for the development of new drugs. **Objectives:** We investigated the pharmacological potential of a series of curcumin-like compounds using two lung adenocarcinoma cell lines as models. **Methods and Results:** Cell viability assay led to the identification of PQM-214 as the hit compound, and other methodologies were employed to investigate the mechanisms underlying its antitumor potential, including cell cycle analysis, mitotic index determination, assessment of clonogenic capacity, senescence-associated β-galactosidase and annexin V assays, quantitative PCR, and Western blot analyses. The mechanism of action of PQM-214 was investigated in A549 cells, revealing that it effectively inhibits cell proliferation by inducing cell cycle arrest, apoptosis, or senescence. Cell cycle key regulators were significantly modulated by PQM-214, with cyclin E2, *MYC*, and *FOXM1* being downregulated, while senescence markers such as cyclin D1, *CDKN1A* (p21), *IL-8*, *TIMP1*, and *TIMP2* were upregulated. Moreover, Western blot results revealed upregulation of cyclin D1 and p21 in PQM-214-treated samples, with a downregulation of cyclin B. **Conclusions**: PQM-214 seems to act on different molecular targets in lung adenocarcinoma cells, inhibiting cell proliferation and inducing apoptosis. Further studies will be conducted to explore whether PQM-214 can also act as a senolytic agent, which would reinforce its anticancer potential.

## 1. Introduction

Lung cancer is the leading cause of cancer-related deaths worldwide [1]. It is a multifactorial and complex disease, classified into small cell lung cancer (SCLC) and non-small cell lung cancer (NSCLC) [2]. NSCLC is the most prevalent type, accounting for 80–85% of lung cancer cases, and lung adenocarcinoma is the most common and lethal NSCLC subtype, responsible for approximately 50% of all lung cancer-related deaths [3,4].

Since most patients are diagnosed at advanced stages of lung cancer, the treatment typically includes chemotherapy, radiofrequency ablation (RFA), radiotherapy, targeted therapies, and immunotherapy [5]. First-line chemotherapy options often involve platinum-based drugs like cisplatin and carboplatin, combined with agents such as paclitaxel, docetaxel, gemcitabine, and others [6]. The discovery of genomic alterations, including mutations in EGFR, ALK, ROS1, RET, BRAF, MET, KRAS, HER2/3, and RAF/MEK, has led to the development of targeted therapies, offering a new approach for advanced NSCLC [7,8,9]. Additionally, immunotherapy employing monoclonal antibodies, such as cetuximab, bevacizumab, nivolumab, and pembrolizumab, is used alone or in combination with other drugs. Furthermore, immune checkpoint inhibitors have emerged as a promising approach for treating lung adenocarcinoma [1]. Despite these advances, innate or acquired resistance, as well as severe side effects, remain a major challenge, limiting clinical efficacy [10]. Thus, the pursuit of new drug prototypes to enhance lung cancer treatment strategies remains ongoing.

Curcumin is a natural polyphenolic compound, primarily found in turmeric (*Curcuma longa*) [11], which has been extensively studied due to its pharmacological properties, including anti-inflammatory, antioxidant, antimicrobial, antitumor, and anti-lipidemic activities [12,13,14,15,16,17]. Regarding its antitumor activity, curcumin can inhibit cancer cell growth, migration, metastasis, and angiogenesis, which are key hallmarks of cancer [18]. In lung cancer, curcumin suppresses cell proliferation, induces apoptosis, and regulates microRNA expression [19,20,21,22,23,24,25,26,27,28,29,30]. Despite its promising antitumor properties, curcumin is not used as a standard anticancer agent due to its poor solubility, low absorption, rapid metabolism, and consequently, low bioavailability [19]. To overcome these limitations, various formulations have been developed to enhance the biological delivery of curcumin, such as curcumin nanoparticles and liposomes [20,21]. Additionally, significant efforts have been made to discover novel synthetic curcumin analogs and curcumin-like compounds with enhanced pharmacokinetic properties [22].

In a previous study, we synthesized a small series of curcumin–resveratrol-based hybrid compounds, incorporating an acyl-hydrazone moiety between the curcumin and resveratrol features, and different substituents on the aromatic ring derived from resveratrol [23]. This series was tested against three cell lines (MCF-7, estrogen-positive breast cancer; A549, lung adenocarcinoma; and HepG2, hepatocellular carcinoma), and the MCF-7 cells showed a higher responsiveness to the PQM-162, which was considered the lead compound (Figure 1). Further, we demonstrated that the antitumor potential of PQM-162 was due to its ability to concomitantly inhibit the expression of critical kinase proteins required for the G2/M transition and mitosis onset, thereby reducing the proliferation of estrogen receptor-positive breast cancer cells [23]. Interestingly, as part of another project focused on neurodegenerative diseases, the first generation of six compounds, including PQM-162, was expanded in the chemical space, leading to a second generation of twenty-three novel curcumin-resveratrol-based *N*-acyl-hydrazones (Figure 1), which were investigated for their neuroprotective effects [24].

Based on previous results, which showed high selectivity of PQM-162 on estrogen-positive breast cancer cells, additional studies were conducted to evaluate the antitumor potential of this second series (29 curcumin-resveratrol-*N*-acyl-hydrazone hybrid compounds). The substances were tested against different tumor cell lines, including MCF-7 cells. Curiously, we found that PQM-214 exhibited potent antiproliferative activity towards lung adenocarcinoma cells (Appendix A), indicating that the structural modification performed in PQM-214 was crucial to its selectivity toward lung adenocarcinoma cells. Here, we describe for the first time a detailed investigation aimed at characterizing the potential mechanisms underlying the effects of PQM-214 on cell proliferation inhibition, which involves inducing cell cycle arrest, senescence, and apoptosis. Importantly, these events were characterized at the molecular level.

## 2. Results

### 2.1. Synthesis of the Target-Compound PQM-214

The curcumin–resveratrol-based N-acyl-hydrazone derivative PQM-214 was synthesized and fully characterized, as previously reported by our group [24]. In brief, the synthetic route comprises a sequential two-step procedure, using commercial ferulic acid as the starting material. In the first step, ferulic acid was converted into the corresponding hydrazide through a reaction with HOBt/EDAC and hydrazine monohydrate in acetonitrile. Next, the feruloyl-hydrazide was reacted with 4-chlorobenzaldehyde under acid catalysis in EtOH, resulting in a 63% yield of the desired N-acyl-hydrazone derivative PQM-214 (Figure 2).

### 2.2. Antiproliferative and Cytotoxic Investigation

Almost all of the twenty-eight compounds tested were able to reduce the viability of adenocarcinoma cells, with PQM-214 standing out as the most active (Appendix A). Its cytotoxic profile was assessed following a 48 h period treatment. Under these experimental conditions, PQM-214 effectively reduced tumor cell viability in a dose-dependent manner compared to non-tumor cells. The IC_50_ (the concentration required to kill 50% of cells) and GI_50_ (the concentration required to reduce cellular proliferation by 50%) values were determined using A549 and H1299 adenocarcinoma cell lines, as well as human cutaneous fibroblasts as a normal cell control (Table 1). Notably, A549 cells exhibited the most pronounced response, making them the optimal choice for subsequent mechanistic investigations into the antitumor potential of PQM-214 against pulmonary adenocarcinoma.

To evaluate the impact of PQM-214 on the proliferative capacity of A549 cells, a clonogenic assay was performed. Cells were treated with 20 µM PQM-214 for 48 h, followed by a 12-day recovery period in fresh medium. Treated cultures exhibited a significant reduction in colony formation compared to controls (Figure 3A), indicating that PQM-214 disrupts the self-renewal and sustained proliferation of A549 cells.

A flow cytometry analysis of cell cycle progression revealed that A549 cells treated with 20 µM PQM-214 for 24 h exhibited a significant accumulation in the G2/M and sub-G1 phases, consistent with mitotic arrest and cell death. These events were accompanied by a marked decrease in G1 and S phase populations (Figure 3B). These findings align with the antiproliferative effect observed for PQM-214.

Consistent with cell cycle arrest, phase-contrast microscopy of A549 cells treated with 20 µM PQM-214 (48 h treatment; 96 h recovery) exhibited a senescent-like morphology, characterized by pronounced cellular enlargement, flattening, and an evident reduction in their proliferative capacity (Figure 3C).

To elucidate the molecular mechanisms underlying G2/M arrest and morphological changes, Western blot analysis was performed. Treatment with 20 µM PQM-214 for 48 h reduced cyclin B1 and c-Myc, while cyclin D1 and p21 were upregulated (Figure 4), suggesting that PQM-214 activates signaling pathways that disrupt mitotic progression while promoting a senescence-like state.

### 2.3. Pro-Senescent Investigation

The senescence-associated β-galactosidase (SA-β-gal) activity assay further corroborated these findings. A549 cells treated with 20 µM PQM-214 for 72 h displayed intense SA-β-gal staining (Figure 5A), a hallmark of senescence. Morphometric analyses confirmed a significant increase in cellular area, a drastic reduction in cell density per field, and a notable elevation in the percentage of SA-β-gal-positive cells (Figure 5B–D).

To delineate the molecular pathways driving senescence, a qPCR analysis of A549 cells treated with 20 µM PQM-214 for 72 h was carried out, which revealed significant modulation of senescence-associated genes. Specifically, a remarkable downregulation of *CCNE2* and *FOXM1* was observed, alongside an upregulation of *CCND1*, *IL-8*, *CDKN1A*, *TIMP1*, and *TIMP2*, as well as suppression of *LMNB1* (Figure 6), suggesting that PQM-214 disrupts mitotic progression and induces senescence-associated secretory phenotype (SASP).

### 2.4. Apoptosis Investigation

In addition to inducing senescence, PQM-214 triggered apoptosis in A549 cells, as demonstrated by a significant increase in annexin V-positive populations following 48 h of treatment (20 µM). Moreover, qPCR analysis revealed an increase in BAX mRNA levels and a decrease in Bcl-2 expression after 72 h (Figure 7), indicating the activation of pro-apoptotic pathways. Altogether, these data establish PQM-214 as a dual-acting agent that induces both senescence and apoptosis, underpinning its potent antitumor efficacy.

## 3. Discussion

In the present study, we investigated the antitumor potential of a series of curcumin–resveratrol-based hybrids. The pharmacophore 4-hydroxy-3-methoxycinnamoyl group from curcumin was linked to a resveratrol-derived functionalized conjugated phenyl ring via an *N*-acyl-hydrazone spacer subunit. This molecular hybridization strategy, combining the structural features of curcumin and resveratrol, while introducing a polar, H-bond donor/acceptor spacer fragment, was employed to explore a broader chemical space in the search for novel ligands with similar or enhanced antitumor activity compared to curcumin and resveratrol. Additionally, this novel molecular architecture could contribute to an improved pharmacokinetic profile, as the incorporation of a hydrazone moiety into chemical scaffolds has been reported to enhance the solubility and oral absorption of drug candidate prototypes [25,26,27,28]. Among the tested compounds, PQM214 was the most active, containing a 4-chloroaryl substituent in the hydrazone moiety. Notably, the cytotoxicity of PQM-214 against adenocarcinoma cells was higher than that of PQM-215 or PQM-216, which also contained a chlorine substituent, but at the *meta*- and *ortho*-positions, respectively. Interestingly, the presence of a 4-methoxy substituent, a small, polar, electron-withdrawing, and H-bond-accepting functionality, in the structure of PQM-162 also conferred enhanced antiproliferative activity against breast cancer cells [23].

The cell viability results revealed that PQM-214 has both cytotoxic and antiproliferative activities on adenocarcinoma cells. These effects were observed in both A549 and H1299 adenocarcinoma cell lines, which have different mutational profiles. Lung cancers are well-known for their high heterogeneity, and genetic differences impact the responsiveness of cancer cells to chemotherapeutic agents. Critical mutations in the *RAS* (rat sarcoma virus) oncogene superfamily, particularly the *KRAS* gene, play a pivotal role in lung adenocarcinoma initiation, progression, and tumor resistance [29,30]. Similarly, mutations in the tumor suppressor gene *TP53* are also associated with cancer development, progression, and metastasis. Since p53 protein is activated under cellular stress conditions, including DNA damage, inactivating mutations of p53 could contribute to genetic instability, cancer progression, and treatment resistance [31]. The A549 cell line contains *KRAS G32S* mutation, while H1299 is null for the *TP53* gene. Our results demonstrated that PQM-214 inhibits the proliferation of both cell lines, and A549 is slightly more sensitive than H1299. Thus, the mechanism of action of PQM-214 was further investigated in A549 cells.

Cell cycle analysis revealed that PQM-214 causes cell cycle arrest at the G2/M transition. The cells exhibited drastic morphological changes in response to the treatment, which became more pronounced with recovery time (96 h). Notably, the morphological features adopted by A549 cells treated with PQM-214 were compatible with a senescent phenotype [32,33]. Then, we investigated the molecular mechanisms underlying cell cycle arrest and the inhibition of proliferation. The Western blot results showed that the expression levels of c-Myc and cyclin E were significantly reduced in response to PQM-214, while the expression of p21 was upregulated. These results may explain the antiproliferative effect of PQM-214 on A549 cells. Moreover, the expression of cyclin D1 was significantly upregulated by the treatment. Therefore, the concomitant upregulation of cyclin D1 and p21 reinforces the hypothesis that PQM-214 induces senescence in A549 cells [34].

Next, we performed functional and molecular investigations to explore the pro-senescent activity of PQM-214. The senescence-associated β-galactosidase assay is a functional assay widely employed to detect senescence induction. As expected, the frequency of cells with lysosomal-β-galactosidase activity was significantly higher in the PQM-214-treated samples compared to controls. Furthermore, we demonstrated that this compound modulates the expression of genes associated with senescence induction. The upregulation of *CDKN1A* (p21), *CCND1* (cyclin D1), *IL-6*, *IL-9*, *TIMP-1*, and *TIMP-2*, along with the concomitant downregulation of *LAMNB1,* is characteristic of senescent cells [34]. Additionally, transcription factors that directly respond to the proliferation stimuli, such as c-Myc and FOXM1, were downregulated, reinforcing the antiproliferative effect of PQM-214 on A549 cells.

Cell senescence is a state of irreversible cell cycle arrest triggered by various stressor factors, including DNA damage, telomere shortening, oxidative stress, and oncogene activation [35]. Senescent cells are commonly characterized not only by durable cell cycle arrest but also by resistance to apoptosis and secretion of several bioactive substances, collectively referred to as senescence-associated secretory phenotype (SASP) [36,37]. Importantly, conventional chemotherapy, cell cycle inhibitors, and epigenetic modulators can induce cell cycle arrest and senescence in tumor cells, a condition known as therapy-induced senescence [33,38]. Theoretically, the induction of senescence in cancer cells would be beneficial due to its negative effect on cell cycle progression. However, studies have shown that the continuous secretion of several pro-inflammatory cytokines, chemokines, growth factors, and extracellular matrix proteins (SASPs) by senescent cells can contribute to tumor resistance and progression [37,39]. Moreover, the senescence induced by cancer therapy may sometimes be reversible, allowing these cells to potentially re-enter the cell cycle, favoring tumor progression and/or recurrence after anticancer therapy [36]. Therefore, it is crucial to have a comprehensive understanding of cellular senescence within the context of cancer biology to develop targeted and effective cancer treatments.

In the present study, we demonstrated that PQM-214 induces cell cycle arrest, senescence, and apoptosis. The expression profiles of apoptosis regulatory genes were modulated by PQM-214. There was a significant increase in the Bax/Bcl-2 ratio, which contributes to triggering intrinsic apoptosis [40,41]. In the literature there are several curcumin-derived and curcumin-related compounds that have been studied as diverse pharmacological targets and for potential clinical development. To the best of our knowledge, compound PQM-214 possesses a genuine structural architecture and was first reported in the literature by our group [23,24]. Compared to other curcumin-related compounds studied both by our group and others, PQM-214 enabled us to explore a broader chemical space. It originated from successive generations of curcumin-based hybrid compounds, which were designed and synthesized by varying substituents at the aryl-hydrazone moiety. This approach led to the identification of compounds with diverse pharmacological profiles, including PQM-214 as a potential antitumor prototype candidate. While curcumin is a well-known compound with poor solubility and significant limitations in terms of bioavailability, PQM-214 and other analogs developed by our group have shown improved solubility and potentially enhanced pharmacokinetic properties, and are currently undergoing detailed investigation. Notably, PQM-214 is a member of a second generation of curcumin-like compounds synthesized by our group. In a previous study, the first series of eight compounds was screened against three cell lines (MCF-7, estrogen-positive breast cancer; A549, lung adenocarcinoma; and HepG2, hepatocellular carcinoma). As a result, MCF-7 cells showed higher responsiveness to compound PQM-162, which was identified as the hit compound of that series, than A549 and HepG2 [23]. Due to the promising results observed for PQM-162, we then designed a second generation of curcumin-based *N*-acyl-aryl hydrazone analogs, aiming to expand the chemical space and to enhance the structure-activity relationship in the search for more active and selective compounds. Among the nineteen new analogs tested against MCF-7 and A549 cell lines, PQM-214 stood out as the most potent compound, showing IC_50_ values of 23.68 μM for A549 (Table 1) and 51.62 for MCF7. Based on these results, which indicated the selectivity of PQM-214 towards lung adenocarcinoma cells and not on breast cancer cells, we decided to include another lung adenocarcinoma cell line (H1299), but with a different genetic background compared to A549. Our findings showed that H1299 cells were less responsive to PQM-214 (IC_50_ = 32.15 μM, Table 1) than A549 cells. However, the effect of this substance was much more pronounced in the H1299 cell line than in MCF-7, indicating that PQM-214 has a certain selectivity toward lung adenocarcinoma cells.

Further studies should be conducted to determine whether PQM-214 can also act as a senolytic agent, which would lead to the clearance of senescent cells [42]. It has already been reported that curcumin and its derivatives, such as EF24, can act as senolytic agents, eliminating non-tumor senescent cells by promoting the degradation of anti-apoptotic proteins via the proteasome [43]. However, the senolytic properties of PQM-214 in lung cancer require further investigations.

## 4. Materials and Methods

### 4.1. Synthesis and Characterization of Compound PQM-214

(*E*)-*N*’-((*E*)-4-chlorobenzylidene)-3-(4-hydroxy-3-methoxyphenyl)-acrylohydrazide:

Ferulic acid (1.68 mmol) was dissolved in 10 mL of acetonitrile, followed by the addition of hydroxybenzotriazole (HOBt, 2.02 mmol) and 1-(3-dimethylaminopropyl)-3-ethyl carbodiimide (EDAC, 2.02 mmol). The solution was stirred at 25 °C for 1.5 h. Then, a solution of hydrazine monohydrate (16.8 mmol) in 5 mL of acetonitrile was prepared and cooled to 0 °C. Then, the reaction mixture was added drop by drop to the cooled hydrazine solution, and the temperature was kept at 0 °C. The reaction was monitored by TLC until its completion, and the solvent was removed under reduced pressure. The crude reaction mixture was resuspended in 4 mL of 5% saturated NaHCO_3_ and the final solution was kept in a freezer for 6 h. The precipitate was collected and washed with cold water to furnish the corresponding feruloyl-hydrazide as a pale solid. The hydrazide intermediate (0.52 mmol) was dissolved in 10 mL of dry ethanol, followed by the addition of 0.62 mmol of 4-chlorobenzaldehyde and a catalytic amount of 40% HCl aq. The reaction mixture was stirred at 25 °C for 24 h. After completion of the reaction (TLC), the solvent was removed under reduced pressure, and cold ethanol was added. The solid formed was collected by filtration to provide the pure compound PQM-214 in a 63% yield, as a yellow solid. Compound PQM-214 was characterized as follows: m.p. 225 °C, purity: 99.9% (HPLC). IR (ATR): ν 3249, 3084, 2985, 1652, 1627, 1586 and 1351 cm^−1^. ^1^H NMR (300 MHz, DMSO- *d_6_*) δ 11.64 and 11.43 (*s*, 1H, NH), 9.56 (*s*, 2H, OH), 8.18 and 8.02 (*s*, 1H, N=CH), 7.77 (*d*, *J* = 8.4 Hz, 2H, Ar-H), 7.72 (*d*, *J* = 8.5 Hz, 2H, Ar-H), 7.58 and 7.52 (*d*, *J* = 16.1 and 15.7 Hz, 1H, HC=CH), 7.36 and 6.50 (*d*, *J* = 16.1 and 15.7 Hz, 1H, HC=CH), 7.49 (*d*, *J* = 8.4 Hz, 4H, Ar-H), 7.27 and 7.18 (*s*, 1H, Ar-H), 7.21 and 7.06 (*d*, *J* = 8.1 Hz, 1H, Ar-H), 6.82 and 6.80 (*d*, *J* = 8.1 Hz, 1H, Ar-H), 3.83 and 3.80 (*s*, 3H, OCH_3_). ^13^C NMR (75 MHz, DMSO-*d*_6_) δ 166.98, 162.44, 149.39, 149.25, 148.29, 145.18, 143.44, 141.93, 141.78, 134.78, 134.54, 133.83, 133.68, 129.35, 129.12, 126.79, 126.55, 122.51, 117.01, 116.17, 113.93, 112.70, 111.38, 56.21 and 55.95. HRMS (ESI) *m*/*z*: Calcd for C_17_H_15_ClN_2_NaO_3_ [M+Na]^+^ 353.0669 found 353.0647 [24].

### 4.2. Cell Cultures

The adenocarcinoma cell lines A549 and H1299 were purchased from the Rio de Janeiro cell Bank, Rio de Janeiro, Brazil, and were tested for mycoplasma contamination periodically. Dermal fibroblasts were also used as a control to represent normal cells in this study. The biological material for primary cultures of fibroblasts was collected following the rules established by the Human Research Ethics Committee (#69453817.9.0000.5142). All cells were cultured in Dulbecco’s Modified Eagle’s Medium (DMEM)/F12 (catalog no. D8900, Sigma-Aldrich, Saint Louis, MO, USA), supplemented with 10% fetal bovine serum (FBS, Cultilab, SP, Brazil). The cell cultures were maintained in an incubator at 37 °C, with a controlled atmosphere (95% air and 5% CO_2_), and subculturing was performed regularly every 2 or 3 days.

### 4.3. Cell Viability Determination

A549 and H1299 cells were seeded on 96-well plates at a density of 5 × 10^3^ cells. Cell viability was assessed using the sulforhodamine B (SRB) colorimetric assay. After 48 h of treatment with PQM-214 at different concentrations, the samples were fixed with 10% (wt/vol) trichloroacetic acid at 4 °C for 1 h and subsequently stained with SRB (0.4% in 1% acetic acid) for 1 h. Excess dye was removed through repeated washes with 1% (vol/vol) acetic acid, and the protein-bound dye was solubilized in 10 mM Tris-base for 30 min. Absorbance was measured at 540 nm, with a reference wavelength of 690 nm, using a microplate reader. The IC_50_ values were calculated using GraphPad Prism^®^ 8.0 software (GraphPad Software, Inc., San Diego, CA, USA). For the determination of GI_50_, cell viability was assessed at two time points, immediately before treatment (T0) and after 48 h of treatment (T48). The percentage growth (PG) was calculated with respect to untreated control cells (C) at each of the drug-concentration levels based on the difference in optical density (OD) according to NCI recommendations [44]. In brief, (OD of treated sample at T48 − DO at T0)/(DO of control at T48 − OD at T0)) × 100.

### 4.4. Clonogenic Assay

A549 cells were seeded at a low density into 35 mm plates (200 cells/plate) and treated with PQM-214 at 20 µM for 48 h. After treatment, the cells were incubated in a drug-free medium for 10 days. The colonies were fixed with methanol for 30 min and stained with 0.5% crystal violet for 5 min. The number of colonies was quantified under a stereomicroscope (Nikon SMZ800, Nikon, Tokyo, Japan), and only colonies containing more than 50 cells were considered for analysis.

### 4.5. Cell Cycle Analysis

Cell cycle progression analysis was performed using flow cytometry. In brief, A549 cells were seeded in 24-well plates at a density of 5 × 10^4^ and treated with PQM-214 at 20 µM for 48 h. Cisplatin was used as a positive control. Following treatment, the cells were collected by enzymatic digestion using a Trypsin-EDTA solution (Sigma Aldrich) and subsequently fixed in 75% ethanol at 4 °C overnight. After treatment with RNase (1.5 mg/mL) and DNA staining using propidium iodide (90 μg/mL, Guava Technologies, Merck Millipore, Burlington, MA, USA) for 1 h at 4 °C, the samples were analyzed by flow cytometry (Guava EasyCyte 8HT).

### 4.6. Gene Expression Analysis Using qPCR

Cells were seeded into 35 mm Petri plates at a density of 2 × 10^5^ cells/plate and treated with PQM-214 at µM for 72 h. Cells were collected by enzymatic digestion, and total RNA was extracted using the RNeasy^®^ Mini kit (Qiagen, Mississauga, ON, Canada). RNA was quantified using a NanoDrop ND 1000 spectrophotometer (NanoDrop Technologies, Wilmington, DE, USA). Subsequently, 1 µg of total RNA was subjected to DNase (1 U; ThermoFisher, Waltham, MA, USA) treatment and then used for reverse transcription (RT) with the High-Capacity cDNA Reverse Transcription Kit (ThermoFisher). Relative mRNA expression of target genes, related to cell cycle regulation, apoptosis, and senescence (Appendix A) was determined as previously described [45] using QuantStudio^TM^ thermal cycler (Applied Biosystems, Foster City, CA, USA) and the amplification protocol of the Power Syber Green Master Mix^®^ kit (Applied Biosystems). The expression values of the target genes were normalized to constitutive β-actin gene expression and the relative abundance of mRNA for each gene was calculated using the ΔΔCt method [46].

### 4.7. Protein Expression Analysis by Western Blot

The expression of proteins associated with cell cycle regulation was evaluated using Western blot analysis. A549 cells were seeded into 100 mm Petri plates (1 × 10^6^ cells/plate) and treated with PQM-214 for 48 h. For protein extraction, RIPA lysis buffer (150 mM NaCl, 1.0% Nonidet P-40, 0.5% deoxycholate, 0.1% SDS and 50 mM Tris pH 8.0) containing both protease and phosphatase inhibitors (Sigma–#P8340) was used. Then, total proteins were quantified (BCA kit, Pierce Biotechnology Inc., Rockford, IL, USA) and the samples, containing 40 μg, were prepared with 4× sample buffer (0.5 M Tris, pH 6.8; 4 mL of glycerol, 10% SDS, 1% bromophenol blue, and 1% beta-mercaptoethanol in deionized water) and heated (100 °C) for 5 min. Protein fractionation was performed on a 12% polyacrylamide mini gel with SDS for 2 h at 100 V, and transferred onto a PVDF membrane (Amersham Pharmacia Biotech, Inc. (Horsham, UK)) for 2 h at 200 mA in transfer buffer (0.025 M Tris, 0.192 M glycine, and 20% methanol in distilled water). The membrane was blocked for 1 h at room temperature with 5% non-fat milk in 0.02 M Tris-buffered saline (TBS) with 0.1% (*v*/*v*) Tween20 (Sigma-Aldrich) and incubated with primary antibodies overnight at 4 °C. Afterwards, the membrane was incubated with a secondary antibody for 2 h at room temperature. Immunoreactive bands were visualized with the ECL Western blotting Detection Kit (Amersham Pharmacia) and the quantification was performed using Image J (Version 1.54p). A list of the antibodies used and their respective dilutions is provided in Appendix A.

### 4.8. Annexin V Assay

Apoptosis was assessed using Annexin-V-PE conjugate and 7-AAD (Merck Millipore, Burlington, MA, USA), following the manufacturer’s instructions. A549 cells were seeded at a density of 1 × 10^5^ cells per 35 mm Petri dish and treated with PQM-214 at 20 µM for 48 h. Cisplatin at 30 µM was used as a positive control. Cells were collected by enzymatic digestion and incubated with Annexin V-PE conjugated and 7-AAD for 30 min in the dark at room temperature. The samples were analyzed by flow cytometry using GuavaSoft 2.7 software (Luminex, Austin, TX, USA).

### 4.9. Cellular Senescence Assay

Cellular Senescence was assessed using Senescence β-Galactosidase Staining Kit (Cell Signaling Technology–#9860, Danvers, MA, USA). A549 cells were seeded at a density of 1 × 10^5^ cells per 35 mm Petri plate and treated with PQM-214 at 10 µM. Cell morphology was monitored at 0, 24, 48, 72 h post-treatment. After 72 h, cells were fixed in 1 mL of fixative solution (provided in the kit) for 15 min. Following fixation, 1 mL of β-galactosidase staining solution was added to each plate, and samples were incubated at 37 °C in a CO_2_-free environment for 7 h. Cells were observed using phase-contrast microscopy, and images were acquired using a blue filter to highlight senescence-associated β-galactosidase activity.

### 4.10. Statistical Analysis

Quantitative data are presented as the mean ± standard deviation (SD) of at least three independent experiments. Significant differences were evidenced according to ANOVA followed by Dunnett’s post-hoc test or Student’s t-test *t*. The software used was GraphPad Prism^®^ 8.0 (GraphPad Software, Inc., San Diego, CA, USA).

## 5. Conclusions

Overall, our findings indicate that the curcumin-like compound PQM-214 can target multiple molecular pathways in lung adenocarcinoma cells, leading to cell cycle arrest, senescence, and apoptosis. Therefore, PQM-214, featuring an innovative curcumin-resveratrol-based *N*-acyl-hydrazone molecular architecture and easy synthetic accessibility, represents a promising multi-target-directed antitumor drug candidate prototype for further investigation and development.

## Figures and Tables

**Figure 1 pharmaceuticals-18-00914-f001:**
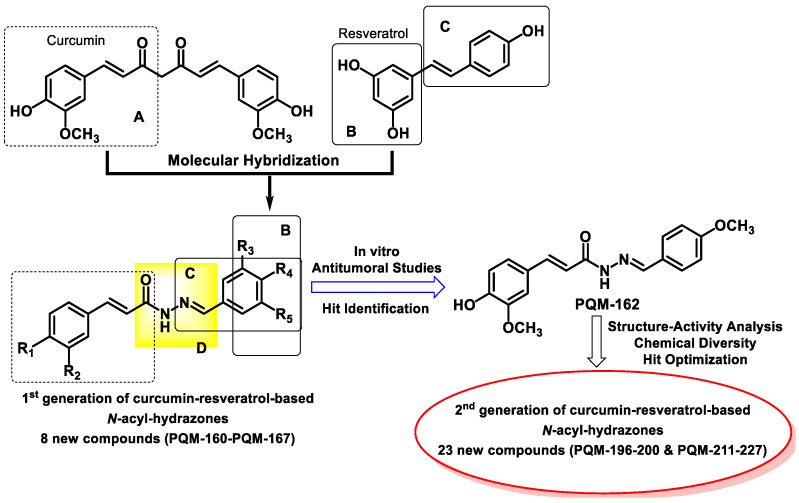
Rational design strategy leading to the first (PQM-160 to PQM-167) and second (PQM-196 to PQM-200 and PQM-211 to PQM-227) generations of hybrid curcumin-resveratrol-*N*-acyl-hydrazone derivatives, and chemical structure of PQM-162, identified as a hit compound for antitumoral activity against breast cancer cells.

**Figure 2 pharmaceuticals-18-00914-f002:**
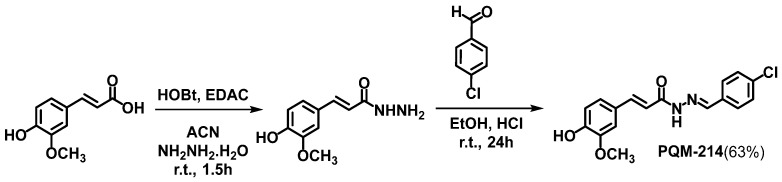
Synthetic route for the curcumin–resveratrol-based *N*-acyl-hydrazone, PQM-214.

**Figure 3 pharmaceuticals-18-00914-f003:**
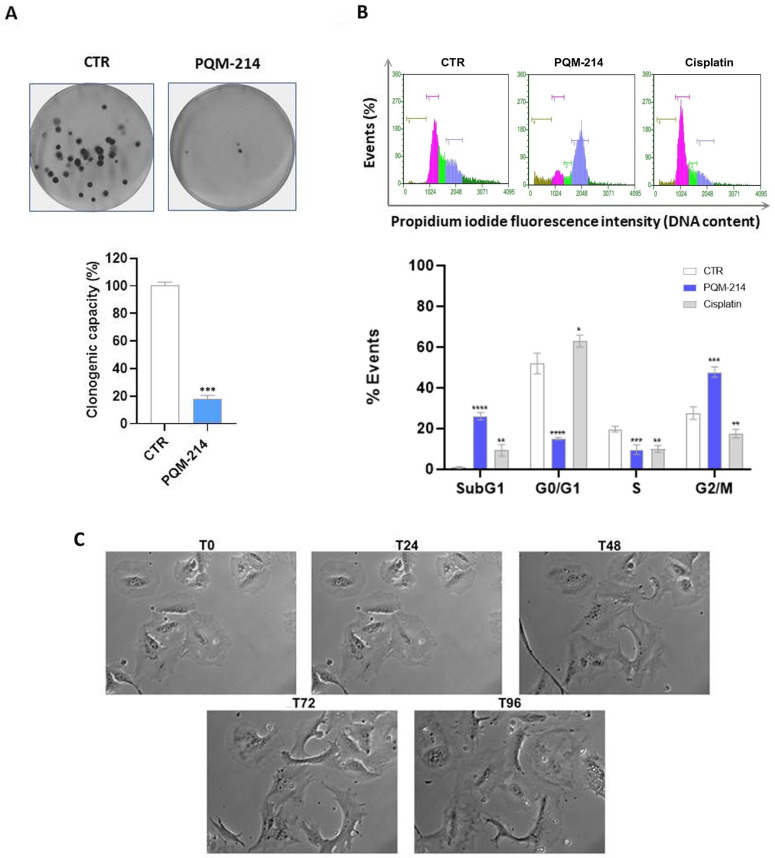
PQM-214 inhibits the proliferation of A549 cells. (**A**) Clonogenic assay. The cells were treated for 48 h with PQM-214 at 20 µM and recovered for 12 days in a fresh medium. (**B**) Cell cycle analysis. The cells were synchronized by removing FBS from the culture medium for 24 h and treating them with PQM-214 at 20 µM for 24 h. Cisplatin at 30 µM was used as a positive control. * *p* ˂ 0.05, ** *p* ˂ 0.01, *** *p* ˂ 0.001, and **** *p* ˂ 0.0001 according to ANOVA followed by Dunnett’s post-hoc test. 1—G1 phase (pink); 2—S phase (light green); 3—G2/M phase (blue) and 4—Sub-G1 phase (brown) (**C**) Morphological features of the cells during recovery period. The cells were treated with PQM-214 at 20 µM for 48 h and recovered for 96 h in a fresh medium. The illustrative images show the same cells after the treatment period. T0 represents the time at which the drug was removed from the sample, and T24, T48, T72, and T96 correspond to successive periods of recovering in drug-free medium (24, 48, 72 and 96 h). Magnification 60×.

**Figure 4 pharmaceuticals-18-00914-f004:**
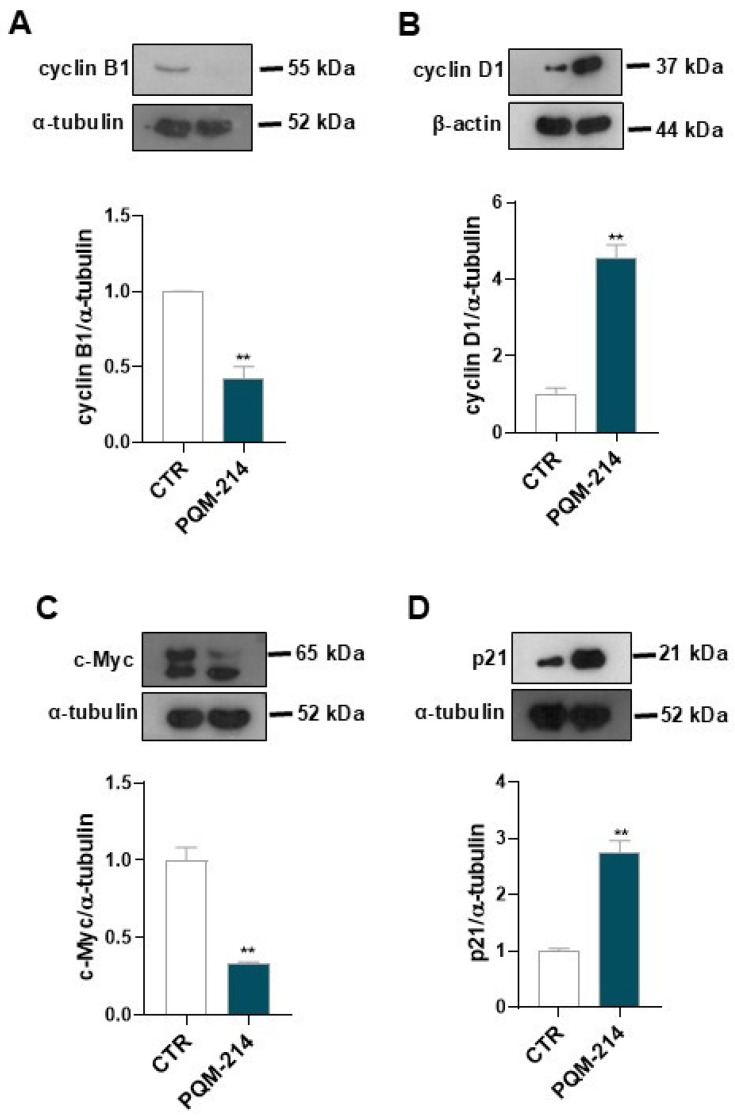
PQM-214 modulates key regulators of cell cycle. Cells were treated with PQM-214 at 20 µM for 48 h and protein expression levels of cyclin B1 (**A**), cyclin D1 (**B**), c-Myc (**C**), and p21 (**D**) were determined by Western blot. ** *p* ˂ 0.01 according to Student’s test.

**Figure 5 pharmaceuticals-18-00914-f005:**
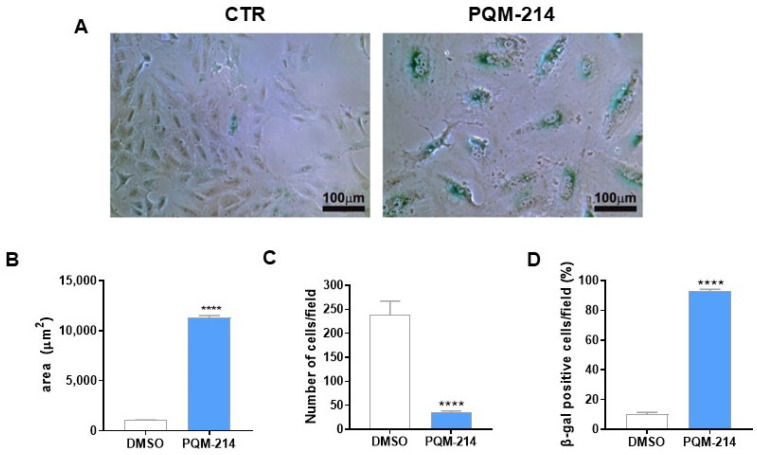
PQM-214 induces senescence in A549 cells. Cells were treated with PQM-214 at 20 µM for 72 h and subjected to a beta-galactosidase senescence assay. (**A**) Illustrative images showing the morphological features of the cells, and characteristic staining pattern for senescent cells are evidence in blue. (**B**) Analysis of the cellular area. (**C**) Determination of number of cells by microscopic field. (**D**) Determination of the frequencies of senescent cells. **** *p* ˂ 0.001 according to Student’s *t*-test.

**Figure 6 pharmaceuticals-18-00914-f006:**
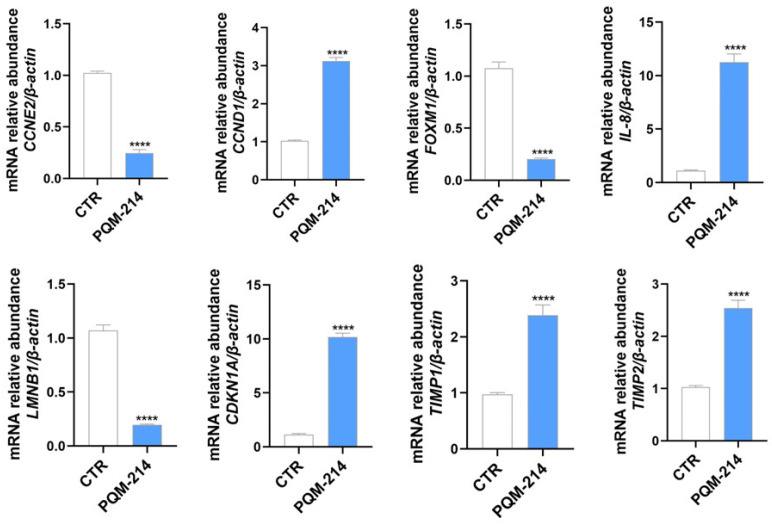
PQM-214 modulates critical markers of senescence. Gene expression profiles determined by qPCR in A549 cells treated with PQM-214 214 at 20 µM for 72 h. **** *p* ˂ 0.001 according to Student’s *t*-test.

**Figure 7 pharmaceuticals-18-00914-f007:**
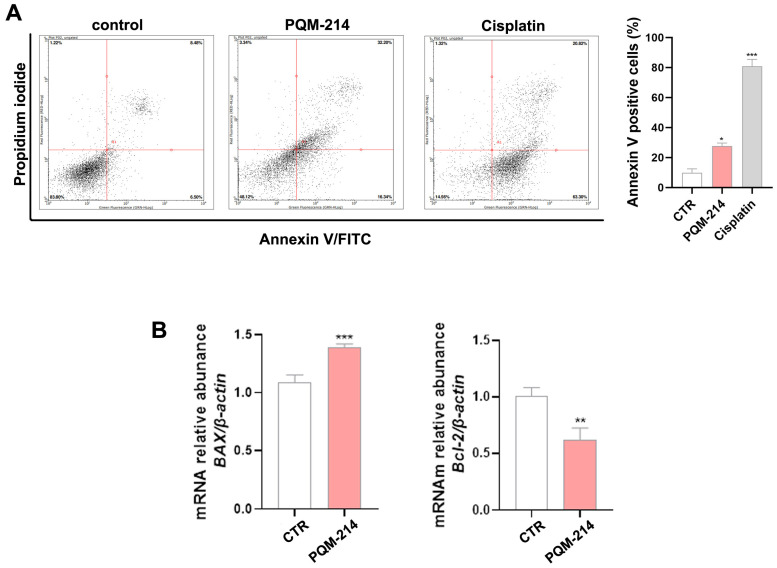
PQM-214 induces apoptosis in A549 cells. Cells were treated with PQM-214 at 20 µM for 48 or 72 h. (**A**) Annexin V assay was performed at 48 h of treatment. Cisplatin at 30 µM was used as a positive control. * *p* ˂ 0.05, *** *p* ˂ 0.001 according to ANOVA followed by Dunnett’s post-hoc test. (**B**) BAX and Bcl-2 expression at mRNA levels were determined by qPCR at 72 h. ** *p* ˂ 0.01, *** *p* ˂ 0.001 according to Student’s *t*-test.

**Table 1 pharmaceuticals-18-00914-t001:** IC_50_ and GI_50_ values (µM) were calculated from viability assays. Cell cultures were treated with PQM-214 at different concentrations for 48 h.

Cell Lines	IC_50_ (PQM-214)	GI_50_ (PQM-214)	IC_50_ (Cisplatin)
A549	23.68 ± 0.66	19.67 ± 0.82	28.13 ± 1.65
H1299	32.15 ± 0.96	23.94 ± 0.83	34.51 ± 1.20
Primary fibroblasts	˃150	˃150	82.95 ± 2.18

## Data Availability

All additional data analyzed during this study can be found in the Appendix A.

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
