# Peer review of "Curcumin-like Compound Inhibits Proliferation of Adenocarcinoma Cells by Inducing Cell Cycle Arrest and Senescence"

_pharmaceuticals, 2025, doi:10.3390/ph18060914_

Round 1

Reviewer 1 Report

Comments and Suggestions for Authors

The manuscript «Curcumin-like compound inhibits proliferation of adenocarcinoma cells by inducing cell cycle arrest and senescence» by Rafael Fonsecaet al. is devoted to the study of the pharmacological potential of a number of curcumin-like compounds using lung adenocarcinoma cell lines (A549 and H1299) as model systems that demonstrate various mutational profiles.

The manuscript is written in good scientific language, describes the results obtained using modern methods and will be of interest to the audience of the journal "Pharmaceuticals", however, there are currently a number of points that need to be considered before further processing of the manuscript.

The manuscript can be published in the journal "Pharmaceuticals" after major revision:

  1. In the section "Materials and methods" the authors did not describe paragraph 4.1.
  2. Lines 308-309. The authors need to provide information about how the GI50 parameter was determined.
  3. In my opinion, the graphs shown in Figure 3 do not carry a valuable semantic load and can be excluded from the text of the manuscript - all the necessary information is present in the tables.
  4. I had an extremely acute feeling of misunderstanding the information presented on lines 260-285. The information presented in these paragraphs contradicts each other and, moreover, contradicts the results obtained by the authors. Throughout the text of the manuscript, the authors emphasize that the PQM-214 molecule leads to the induction of aging. In turn, in the text presented on the above lines, the authors for some reason emphasize that aging induced by therapy plays a contradictory role, and PQM-214 should be further investigated as a senolytic agent. How could the authors come to this conclusion? In their work, the authors prove the radically opposite effect of PQM-214 - the induction of senescent cell formation! In my opinion, it is absolutely pointless to consider PQM-214 as a senolytic - the authors have proven its TIS effect in their work.

Of course, the text describing the contradictory effects of aging-related therapy should be included in the discussion section (the authors should not delete it) - this highlights the range of diverse effects of curcumin-like molecules. However, in my opinion, the authors should introduce 2-3 phrases that could help the reader grasp exactly the above-mentioned idea, for example, "despite the fact that contradictory data are described in the literature for curcumin-like compounds concerning in some cases the stimulation of the formation of senescent cells [], and in others their elimination from the body [], The design and synthesis of the molecules described in our work allowed us to obtain compounds with the potential to be considered as molecules with a pronounced ability to exert a TIS effect." Moreover, I would like to recommend that the authors cite more modern works that emphasize the paradoxical effects of curcumin-like molecules, in particular:

- Yuan, C.; Fan, R.; Zhu, K.; Wang, Y.; Xie, W.; Liang, Y. Curcumin induces ferroptosis and apoptosis in osteosarcoma cells by regulating Nrf2/GPX4 signaling pathway. Exp. Biol. Med. 2023, 248, 2183–2197.

- Li, W.; He, Y.; Zhang, R.; Zheng, G.; Zhou, D. The curcumin analog EF24 is a novel senolytic agent. Aging (Albany NY) 2019, 11, 771–782.

- Aleksandrova Y, Neganova M. Antioxidant Senotherapy by Natural Compounds: A Beneficial Partner in Cancer Treatment. Antioxidants. 2025; 14(2):199.

- Singh, S.; Barnes, C.A.; D’Souza, J.S.; Hosur, R.V.; Mishra, P. Curcumin, a potential initiator of apoptosis via direct interactions with Bcl-xL and Bid. Proteins 2022, 90, 455–464.

- Yang, L.; Shi, J.; Wang, X.; Zhang, R. Curcumin Alleviates D-Galactose-Induced Cardiomyocyte Senescence by Promoting Autophagy via the SIRT1/AMPK/mTOR Pathway. Evid. Based Complement. Altern. Med. 2022, 2022, 2990843.

- Garcia-Trejo, S.S.; Gomez-Sierra, T.; Eugenio-Perez, D.; Medina-Campos, O.N.; Pedraza-Chaverri, J. Protective Effect of Curcumin on D-Galactose-Induced Senescence and Oxidative Stress in LLC-PK1 and HK-2 Cells. Antioxidants 2024, 13, 415.

I would like to wish the authors success in further improving the text of the manuscript! I hope that my comments will be useful to them.

Author Response

The manuscript «Curcumin-like compound inhibits proliferation of adenocarcinoma cells by inducing cell cycle arrest and senescence» by Rafael Fonseca et al. is devoted to the study of the pharmacological potential of a number of curcumin-like compounds using lung adenocarcinoma cell lines (A549 and H1299) as model systems that demonstrate various mutational profiles.

The manuscript is written in good scientific language, describes the results obtained using modern methods and will be of interest to the audience of the journal "Pharmaceuticals", however, there are currently a number of points that need to be considered before further processing of the manuscript.

The manuscript can be published in the journal "Pharmaceuticals" after major revision:

  1. In the section "Materials and methods" the authors did not describe paragraph 4.1.

Reply: Our apologies for the oversight. We have included the information on the synthesis and characterization of the compounds in the revised version of the manuscript. Thank you for your valuable comment.

  1. Lines 308-309. The authors need to provide information about how the GI50 parameter was determined.

Reply: Thank you for this comment. We added more information in the Methods section to clarify how GI50 was determined.

  1. In my opinion, the graphs shown in Figure 3 do not carry a valuable semantic load and can be excluded from the text of the manuscript - all the necessary information is present in the tables.

Reply: The figure 3 was excluded from the manuscript as requested. 

  1. I had an extremely acute feeling of misunderstanding the information presented on lines 260-285. The information presented in these paragraphs contradicts each other and, moreover, contradicts the results obtained by the authors. Throughout the text of the manuscript, the authors emphasize that the PQM-214 molecule leads to the induction of aging. In turn, in the text presented on the above lines, the authors for some reason emphasize that aging induced by therapy plays a contradictory role, and PQM-214 should be further investigated as a senolytic agent. How could the authors come to this conclusion? In their work, the authors prove the radically opposite effect of PQM-214 - the induction of senescent cell formation! In my opinion, it is absolutely pointless to consider PQM-214 as a senolytic - the authors have proven its TIS effect in their work.

Of course, the text describing the contradictory effects of aging-related therapy should be included in the discussion section (the authors should not delete it) - this highlights the range of diverse effects of curcumin-like molecules. However, in my opinion, the authors should introduce 2-3 phrases that could help the reader grasp exactly the above-mentioned idea, for example, "despite the fact that contradictory data are described in the literature for curcumin-like compounds concerning in some cases the stimulation of the formation of senescent cells [], and in others their elimination from the body [], The design and synthesis of the molecules described in our work allowed us to obtain compounds with the potential to be considered as molecules with a pronounced ability to exert a TIS effect." Moreover, I would like to recommend that the authors cite more modern works that emphasize the paradoxical effects of curcumin-like molecules, in particular:

- Yuan, C.; Fan, R.; Zhu, K.; Wang, Y.; Xie, W.; Liang, Y. Curcumin induces ferroptosis and apoptosis in osteosarcoma cells by regulating Nrf2/GPX4 signaling pathway. Exp. Biol. Med. 2023, 248, 2183–2197.

- Li, W.; He, Y.; Zhang, R.; Zheng, G.; Zhou, D. The curcumin analog EF24 is a novel senolytic agent. Aging (Albany NY) 2019, 11, 771–782.

- Aleksandrova Y, Neganova M. Antioxidant Senotherapy by Natural Compounds: A Beneficial Partner in Cancer Treatment. Antioxidants. 2025; 14(2):199.

- Singh, S.; Barnes, C.A.; D’Souza, J.S.; Hosur, R.V.; Mishra, P. Curcumin, a potential initiator of apoptosis via direct interactions with Bcl-xL and Bid. Proteins 2022, 90, 455–464.

- Yang, L.; Shi, J.; Wang, X.; Zhang, R. Curcumin Alleviates D-Galactose-Induced Cardiomyocyte Senescence by Promoting Autophagy via the SIRT1/AMPK/mTOR Pathway. Evid. Based Complement. Altern. Med. 2022, 2022, 2990843.

- Garcia-Trejo, S.S.; Gomez-Sierra, T.; Eugenio-Perez, D.; Medina-Campos, O.N.; Pedraza-Chaverri, J. Protective Effect of Curcumin on D-Galactose-Induced Senescence and Oxidative Stress in LLC-PK1 and HK-2 Cells. Antioxidants 2024, 13, 415.

Reply: Thank you for this comment and for the opportunity to discuss this issue. Indeed, the original text was confusing. To address this, we rewrote the last paragraph of the Discussion section to clarify the information. Actually, PQM-214 has antitumor potential by inhibiting cell cycle progression and promoting and apoptosis.  However, this compound also induces senescence in A549 cells, which could be problematic in the tumor microenvironment whether the senescent cells are not removed. Recent studies have explored the therapeutic effectiveness of combining the pro-senescent agents with senolytics, the latter being used to eliminate senescent cells. Taken together, our findings allow us to hypothesize that PQM-214 may act through a dual mechanism, i.e., inducing senescence and subsequently eliminating these cells by acting as a senolytic agent, which could enhance its antitumor activity. However, further studies are required to confirm this hypothesis.  

We hope the revised text clearly conveys the ideas contextualized above.

Thank you for kindly suggesting additional references. However, Reviewer #2 recommended drastically reducing the reference list. Thus, we included the study by Li et al. (2019), considering that the study involves the senolytic effect of a curcumin analog (EF24) on non-tumor cells. This reference is cited as [44].

I would like to wish the authors success in further improving the text of the manuscript! I hope that my comments will be useful to them.

Reply: Thank you for your valuable contributions, which were crucial to improving the quality of our manuscript.

Reviewer 2 Report

Comments and Suggestions for Authors

The conclusion of the abstract section needs improvement. In its current way, the conclusion does not reflect the information obtained in this work.

Figure 3 and Table 1 present similar information. Figure 3 should be erased from the manuscript.

There are 55 references in the manuscript. I recommend reducing the number of references to 40.

Table 1. A reference drug of cytotoxicity is necessary in this assay.

The term antitumoral refers to in vivo studies; the authors performed an in vitro study.

It is necessary to include a reference drug for the cell cycle analysis, the clonogenic assay, and the annexin V assay.

Figure 4c. What is the justification for showing the cell's morphology?

The manuscript should include an explanation for the upcoming comments.

a) Are the cells from the Rio de Janeiro cell bank sourced from the ATCC? How does this cell bank work? Is it known at an international level?

b) How were the primary fibroblasts obtained?

c) How were the IC50 and GI50 values calculated?

d) Is there a justification for using only two cancer cell lines??

e) Are there any other related compounds reported in the literature like the one described in this work? What are the advantages of this curcumin-like compound?

Author Response

  1. The conclusion of the abstract section needs improvement. In its current way, the conclusion does not reflect the information obtained in this work.

Reply: Done.

  1. Figure 3 and Table 1 present similar information. Figure 3 should be erased from the manuscript.

Reply: Figure 3 was excluded from the manuscript as requested.

  1. There are 55 references in the manuscript. I recommend reducing the number of references to 40.

Reply: We revised the citations. Unfortunately, it was not possible to fully address this recommendation. However, the reference list was reduced as requested.

  1. Table 1. A reference drug of cytotoxicity is necessary in this assay.

Reply: Although our principal objective is not to compare the potency of the tested prototype with that of other chemotherapeutics, we included a reference drug, as requested.

  1. The term antitumoral refers to in vivo studies; the authors performed an in vitro study.

Reply: We agree completely with this comment. It is very important to use appropriate nomenclature for in vitro and in vivo experiments. However, we performed a pool of the results that allowed us to state that PQM-214 has antitumor potential. Thus, we would like to maintain this narrative.

  1. It is necessary to include a reference drug for the cell cycle analysis, the clonogenic assay, and the annexin V assay.

Reply: We agree that including a reference drug in these experimental approaches could be useful for validation of acquired conditions and analyses. However, these approaches are well-standardized and routinely employed in our laboratory. Additionally, we applied complementary methodologies to demonstrate the antiproliferative (cell viability, clonogenic assay, and cell cycle analysis) and pro-apoptotic (Annexin V staining and determination of pro-apoptotic BAX and the anti-apoptotic Bcl-2 genes expression) activities of PQM-214, with consistent and convergent results. The inclusion of a reference drug would provide information regarding the potency of PQM-214, however, this was not the primary focus of the present study.

Figure 4c. What is the justification for showing the cell's morphology?

Reply: Thank you for the opportunity to discuss this issue. These images were captured from the same microscope field and, therefore, show the same cells throughout the recovery period. The image set demonstrates that the cells did not undergo mitosis over the 96-hour period and that the senescent phenotype became more prominent in the post-exposure period to PQM-214.

  1. The manuscript should include an explanation for the upcoming comments.
  2. a) Are the cells from the Rio de Janeiro cell bank sourced from the ATCC? How does this cell bank work? Is it known at an international level?

Reply: The Rio de Janeiro Cell Bank is a research institute in Brazil licensed to import cell cultures from ATCC or other international sources. Many scientific papers have been published using cell lines obtained from the Rio de Janeiro Cell Bank, including articles in journals such as Pharmaceuticals, Biomedicines, and Molecules.

Melo ML, Fonseca R, Pauli F, Zavan B, Hanemann JAC, Miyazawa M, Caixeta ES, Nacif JLM, Aissa AF, Barreiro EJ, Ionta M. N-acylhydrazone derivative modulates cell cycle regulators promoting mitosis arrest and apoptosis in estrogen positive MCF-7 breast cancer cells. Toxicol In Vitro. 2023 Dec; 93:105686. doi: 10.1016/j.tiv.2023.105686. Epub 2023 Aug 29. PMID: 37652252.

Andrade AAR, Pauli F, Pressete CG, Zavan B, Hanemann JAC, Miyazawa M, Fonseca R, Caixeta ES, Nacif JLM, Aissa AF, Barreiro EJ, Ionta M. Antiproliferative Activity of N-Acylhydrazone Derivative on Hepatocellular Carcinoma Cells Involves Transcriptional Regulation of Genes Required for G2/M Transition. Biomedicines. 2024 Apr 18;12(4):892. doi: 10.3390/biomedicines12040892. PMID: 38672246; PMCID: PMC11048582.

Ionta M, Ferreira-Silva GA, Niero EL, Costa ÉD, Martens AA, Rosa W, Soares MG, Machado-Santelli GM, Lago JH, Santos MH. 7-Epiclusianone, a Benzophenone Extracted from Garcinia brasiliensis (Clusiaceae), Induces Cell Cycle Arrest in G1/S Transition in A549 Cells. Molecules. 2015 Jul 15;20(7):12804-16. doi: 10.3390/molecules200712804. PMID: 26184153; PMCID: PMC6332126.

Pressete CG, Viegas FPD, Campos TG, Caixeta ES, Hanemann JAC, Ferreira-Silva GÁ, Zavan B, Aissa AF, Miyazawa M, Viegas C Jr, Ionta M. Piperine-Chlorogenic Acid Hybrid Inhibits the Proliferation of the SK-MEL-147 Melanoma Cells by Modulating Mitotic Kinases. Pharmaceuticals (Basel). 2023 Jan 19;16(2):145. doi: 10.3390/ph16020145. PMID: 37259298; PMCID: PMC9965075.

  1. b) How were the primary fibroblasts obtained?

Reply: Primary human dermal fibroblast cultures were established from skin samples obtained during elective surgeries, in accordance with the guidelines set by the Human Research Ethics Committee (approval number #69453817.9.0000.5142). Our university has a formal agreement with a public hospital in the city of Alfenas to collect both normal and tumor tissue samples. These primary cultures are available in the university's biobank for oncology research.

  1. c) How were the IC50 and GI50 values calculated?

Reply: We determined the IC50 (half maximal inhibitory concentration) and GI50 (50% growth inhibition) values, which are parameters used to evaluate the cytotoxic and cytostatic effects of drugs, respectively, using GraphPad Prism® 8.0. For IC₅₀ determination, cell cultures were treated with at least five different concentrations of PQM-214 for 48 hours, and cell viability was assessed using a microplate reader. For GI₅₀ determination, cell viability was measured at two-time points—immediately before treatment (T0) and after 48 hours of treatment (T48)—allowing us to calculate cell growth over the 48-hour period. Thank you for this comment. We included more information about these issues in the Methods section.

  1. d) Is there a justification for using only two cancer cell lines??

Reply: A panel of cancer cell lines is commonly used for screening promising drug candidate prototypes. However, in this study, we are at a subsequent stage, where our goal is to demonstrate the antitumor potential of the previously selected compound against a specific type of cancer, lung adenocarcinoma in this case, and to characterize its mechanism of action. From this perspective, we included two cell lines with different genetic backgrounds.

  1. e) Are there any other related compounds reported in the literature like the one described in this work? What are the advantages of this curcumin-like compound?

Reply: To the best of our knowledge, the series from which compound PQM-214 originated is genuine and was first reported in the literature by our group, in two different publications cited in the manuscript. The advantage of this series of compounds, and PQM-214 in particular, is that it allowed us to explore a broader chemical space. By synthesizing a variety of analogues with different substituents, we were able to identify compounds with diverse pharmacological profiles, including PQM-214. As for curcumin, it is a well-known poorly soluble compound with significant limitations in terms of bioavailability. In contrast, PQM-214 and other curcumin-based analogues developed by our group showed improved solubility and potentially better pharmacokinetic properties, which are currently under detailed investigation.

Round 2

Reviewer 2 Report

Comments and Suggestions for Authors

Even if the authors state that these assays are well-standardized and routinely employed, it is necessary to include a reference drug for the cell cycle analysis, the clonogenic assay, and the annexin V assay. To ensure proper controls in each experiment, the results must display them clearly.

In my opinion, the cell's morphology is not relevant to be presented in this work. The authors are presenting other results.

The justification for using only two cancer cell lines is not clear.

The explanations of the following comments must be briefly included in the manuscript.

a) Are the cells from the Rio de Janeiro cell bank sourced from the ATCC? How does this cell bank work? Is it known at an international level?

b) How were the primary fibroblasts obtained?

c) Are there any other related compounds reported in the literature like the one described in this work? What are the advantages of this curcumin-like compound?

Author Response

  1. Even if the authors state that these assays are well-standardized and routinely employed, it is necessary to include a reference drug for the cell cycle analysis, the clonogenic assay, and the annexin V assay. To ensure proper controls in each experiment, the results must display them clearly.

Reply: We included additional experiments as required. It is important to clarify that the experiments were performed again to incorporate an additional sample into our analyses: a positive control (cisplatin).

In my opinion, the cell's morphology is not relevant to be presented in this work. The authors are presenting other results.

Reply: We wish to retain the morphology images to demonstrate that, up to 96 hours post-treatment, the cells did not undergo mitosis. Furthermore, the cells exhibited increased size and a flattened shape, which are characteristic features of senescent cells

The justification for using only two cancer cell lines is not clear.

Reply: Thank you for the opportunity to clarify why only two tumor cell lines were used. We apologize for not addressing this point in our initial response. The primary objective of this study was to evaluate the effects of the synthesized compounds on a specific subtype of lung cancer (lung adenocarcinoma), which is both highly prevalent and associated with high mortality, particularly in men. The two cell lines used in our experiments are both derived from lung adenocarcinoma but exhibit distinct mutation profiles. We observed that A549 cells were more responsive to treatment than H1299 cells. Therefore, A549 cells were selected to explore the mechanism of action of PQM-214 on adenocarcinoma cells.

The explanations of the following comments must be briefly included in the manuscript.

  1. a) Are the cells from the Rio de Janeiro cell bank sourced from the ATCC? How does this cell bank work? Is it known at an international level?

Reply: The Rio de Janeiro Cell Bank has the license to import the cell lines from ATCC and to sell them in our country. As mentioned in the first round, our research group has published many articles where cell lines purchased from the Rio de Janeiro Cell Bank were used as experimental models.

Melo ML, Fonseca R, Pauli F, Zavan B, Hanemann JAC, Miyazawa M, Caixeta ES, Nacif JLM, Aissa AF, Barreiro EJ, Ionta M. N-acylhydrazone derivative modulates cell cycle regulators promoting mitosis arrest and apoptosis in estrogen positive MCF-7 breast cancer cells. Toxicol In Vitro. 2023 Dec;93:105686. doi: 10.1016/j.tiv.2023.105686. Epub 2023 Aug 29. PMID: 37652252.

Andrade AAR, Pauli F, Pressete CG, Zavan B, Hanemann JAC, Miyazawa M, Fonseca R, Caixeta ES, Nacif JLM, Aissa AF, Barreiro EJ, Ionta M. Antiproliferative Activity of N-Acylhydrazone Derivative on Hepatocellular Carcinoma Cells Involves Transcriptional Regulation of Genes Required for G2/M Transition. Biomedicines. 2024 Apr 18;12(4):892. doi: 10.3390/biomedicines12040892. PMID: 38672246; PMCID: PMC11048582.

Ionta M, Ferreira-Silva GA, Niero EL, Costa ÉD, Martens AA, Rosa W, Soares MG, Machado-Santelli GM, Lago JH, Santos MH. 7-Epiclusianone, a Benzophenone Extracted from Garcinia brasiliensis (Clusiaceae), Induces Cell Cycle Arrest in G1/S Transition in A549 Cells. Molecules. 2015 Jul 15;20(7):12804-16. doi: 10.3390/molecules200712804. PMID: 26184153; PMCID: PMC6332126.

Pressete CG, Viegas FPD, Campos TG, Caixeta ES, Hanemann JAC, Ferreira-Silva GÁ, Zavan B, Aissa AF, Miyazawa M, Viegas C Jr, Ionta M. Piperine-Chlorogenic Acid Hybrid Inhibits the Proliferation of the SK-MEL-147 Melanoma Cells by Modulating Mitotic Kinases. Pharmaceuticals (Basel). 2023 Jan 19;16(2):145. doi: 10.3390/ph16020145. PMID: 37259298; PMCID: PMC9965075.

  1. b) How were the primary fibroblasts obtained?

Reply: As stated previously, the primary human dermal fibroblast cultures were established from skin samples, obtained during elective surgeries, in accordance with the guidelines set by the Human Research Ethics Committee (approval number #69453817.9.0000.5142). We added this information to the manuscript as requested.

  1. c) Are there any other related compounds reported in the literature like the one described in this work? What are the advantages of this curcumin-like compound?

Reply: To the best of our knowledge, the series from which compound PQM-214 originated is genuine and was first reported in the literature by our group, in two different publications cited in the manuscript. The advantage of this series of compounds, and in particular PQM-214, is that it allowed us to explore a broader chemical space. By synthesizing a variety of analogues with different substituents, we could identify compounds with diverse pharmacological profiles, including PQM-214. As for curcumin, it is a well-known poorly soluble compound with significant limitations in terms of bioavailability. In contrast, PQM-214 and other curcumin-based analogues developed by our group showed improved solubility and potentially better pharmacokinetic properties, which are currently under detailed investigation.

Round 3

Reviewer 2 Report

Comments and Suggestions for Authors

The justification for using only two cancer cell lines is still not clear. The author's response is based on the mortality and prevalence of cancer. However, there are other types of cancer with similar or more prevalence and mortality than lung cancer and prostate cancer. I encourage the authors to use at least 2-3 cell lines to justify the use of these compounds for these two types of cancer.

Are there any other related compounds reported in the literature like the one described in this work? What are the advantages of this curcumin-like compound? The authors' reply should be incorporated into the manuscript.

Author Response

  1. The justification for using only two cancer cell lines is still not clear. The author's response is based on the mortality and prevalence of cancer. However, there are other types of cancer with similar or more prevalence and mortality than lung cancer and prostate cancer. I encourage the authors to use at least 2-3 cell lines to justify the use of these compounds for these two types of cancer.

Reply: Thank you for the opportunity to discuss this issue again.

PQM-214 is a member of the second generation of curcumin-like compounds synthesized by our group. It is essential to claim that the first series was assayed against three cell lines (MCF-7, estrogen-positive breast cancer; A549, lung adenocarcinoma; and HepG2, hepatocellular carcinoma), and MCF-7 cells showed higher responsiveness to the PQM-162 (lead compound of the series) than A549 and HepG2 (Freitas et al., 2018). When the second curcumin-like series was obtained, we tested all substances against MCF-7 and A549 cell lines. The IC50 values of PQM-214 were 23.68 ± 0.66 for A549 (as shown in Table 1 of the manuscript) and 51.62 ± 0.64 for MCF7 (data not shown). Based on these results, which indicate the selectivity of PQM-214 on lung adenocarcinoma cells and not on breast cancer cells, we decided to include another lung adenocarcinoma cell line (H1299), but with a different genetic background compared to A549. Our findings showed that H1299 cells were less responsive to PQM-214 (IC50 = 32.15 ± 0.96, see Table 1 of the manuscript) than A549 cells. However, the effect of this substance was much more pronounced in the H1299 cell line than in MCF-7, indicating that PQM-214 has a certain selectivity toward lung adenocarcinoma cells. Thus, we decided to focus our study on the lung adenocarcinoma model and used the A549 cell line to demonstrate at the molecular level the mechanism of action of PQM-214. Previous studies and the history of rational planning of PQM-214 were contextualized in the Introduction section. We would like to reinforce that our objective in this study was not to conduct an extensive screening, but to demonstrate the mechanism of action of PQM-214.

Round 4

Reviewer 2 Report

Comments and Suggestions for Authors

The authors reponse regarding the justification for using only two cancer cell lines should be incorporated into the manuscript.

Author Response

The authors would like to thank the Editor and Reviewers for their comments and criticism, which allowed us to improve the quality of the manuscript.

Reviewer #2:

  1. The authors response regarding the justification for using only two cancer cell lines should be incorporated into the manuscript.

Reply: We would like to thank the reviewer for his/her time, patience, and valuable suggestions.

We contextualized in the Introduction section of the scientific rational design to justify why we used two cell lines derived from lung adenocarcinoma cells.

Please, see the modifications performed in the revised manuscript (pages 2-3, lines 91-104 and lines 102-118). In the discussion section, a second detailed sentence was introduced, describing the sequential generation of a new series of compounds, which led to the identification of PQM-214 as the most promising analogue.
